# Compensated Integrated Gradients for Reliable Explanation of Electroencephalogram Signal Classification

**DOI:** 10.3390/brainsci12070849

**Published:** 2022-06-28

**Authors:** Yuji Kawai, Kazuki Tachikawa, Jihoon Park, Minoru Asada

**Affiliations:** 1Symbiotic Intelligent Systems Research Center, Institute for Open and Transdisciplinary Research Initiatives, Osaka University, Osaka 565-0871, Japan; jihoon.park@otri.osaka-u.ac.jp (J.P.); asada@otri.osaka-u.ac.jp (M.A.); 2Graduate School of Engineering, Osaka University, Osaka 565-0871, Japan; kazuki.tachikawa@ams.eng.osaka-u.ac.jp; 3Center for Information and Neural Networks, National Institute of Information and Communications Technology, Osaka 565-0871, Japan; 4International Professional University of Technology in Osaka, Osaka 530-0001, Japan; 5Academy of Emerging Science, Chubu University, Aichi 487-8501, Japan

**Keywords:** integrated gradients, deep learning, explainability, Shapley sampling, EEG signal classification

## Abstract

The integrated gradients (IG) method is widely used to evaluate the extent to which each input feature contributes to the classification using a deep learning model because it theoretically satisfies the desired properties to fairly attribute the contributions to the classification. However, this approach requires an appropriate baseline to do so. In this study, we propose a compensated IG method that does not require a baseline, which compensates the contributions calculated using the IG method at an arbitrary baseline by using an example of the Shapley sampling value. We prove that the proposed approach can compute the contributions to the classification results reliably if the processes of each input feature in a classifier are independent of one another and the parameterization of each process is identical, as in shared weights in convolutional neural networks. Using three datasets on electroencephalogram recordings, we experimentally demonstrate that the contributions obtained by the proposed compensated IG method are more reliable than those obtained using the original IG method and that its computational complexity is much lower than that of the Shapley sampling method.

## 1. Introduction

Deep learning methods have shown considerable promise for a wide range of applications, such as image recognition, natural language processing, and speech recognition. They have even attracted attention for the classification of diseases, particularly in seizure detection [1,2] and the evaluation of physiological states of the brain and body [3,4] from raw electroencephalogram (EEG) signals [5]. Recently, deep learning techniques have been applied to the analysis of EEG signals to screen for several psychiatric disorders (e.g., [6,7]). However, understanding how deep learning classifiers work is difficult for users, because their decisions are not fully explained. Therefore, there is a strong demand for explainable or interpretable classifiers with high performance for use in medical applications [8,9,10]. For example, in medical decision support systems, explanations of the output of a machine-learning-based classifier would enable clinicians to corroborate its classification results against existing medical knowledge and possibly discover previously unknown features of a disease. This would also improve both clinicians’ and patients’ trust in deep learning methods.

Many methods have been proposed to visualize the separate contributions of input features to classification results [4,11,12,13,14,15]. Such saliency methods highlight input subspaces (e.g., pixels in the case of image recognition) that are important in identifying a classification label of a given data sample. Among these approaches, integrated gradients (IG) [13] and Shapley sampling (hereafter, SS) [11,16] have been shown to be theoretically superior to other methods because they satisfy the desired properties for the fair attribution of contributions to the classification [13,15]. Although the IG method is computationally efficient, it requires an appropriate baseline or reference point to determine reliable contributions to the classification. The baseline is an input that is assumed to not include any features, and it is set empirically (usually, the zero point) as no formal methods are available to find an appropriate baseline. Setting an inappropriate baseline may undermine the reliability of the attributions [17]. An appropriate baseline depends on the domain, task, and classifier, and they are currently only set on a case-by-case basis. In contrast, the SS method does not require a baseline [11]; however, its computational cost is extremely high [15].

Explanation techniques have been applied to EEG signal classification. Schirrmeister et al. [4] proposed an input perturbation network-prediction correlation map to interpret how convolutional neural networks (CNNs) classify EEG signals. They added noise to each input feature and computed the correlations between the noise levels and the changes in classifier output values. Highly correlated input features were assumed to contribute more to producing the output. Chen et al. [7] exploited the more sophisticated gradient-weighted class activation mapping (Grad-CAM) explanation method in which output class information is backpropagated to an input feature space through CNN filters [12]. However, these methods are not guaranteed to satisfy the desired properties for a good explanation. Therefore, we previously used the SS method to reliably explain a CNN model’s classification of EEG signals [18]. However, the SS method has an extremely high computational cost for a high-dimensional input space, such as EEG signals. In contrast, the IG method has a lower computational cost, but it suffers from the limitation of requiring a baseline to be set for each usage.

In this study, we propose a method to compensate the input contributions to the classification results obtained using the IG method with an arbitrary baseline by using an SS contribution. The proposed method satisfies the same desired properties as the IG method with an appropriate baseline under specific constraints on the classifier, i.e., a linear combination of each filter output. We experimentally evaluated the reliability and computational complexities of the proposed compensated IG method on three different EEG datasets. This method calculates the contributions of electrodes to the classification of a brain state or disease for each data sample. We averaged the contributions over some data samples and visualized the scalp areas that were important for the classification.

Below, in Section 2, we review the existing methods, namely, the Shapley value, SS, and IG, which satisfy the desired properties for a reliable explanation. Of note, the IG method can satisfy the desired properties only with a properly set baseline. We then describe the proposed method for compensating IG with an arbitrary baseline in Section 3. Section 4 describes an experimental evaluation of the reliability of the proposed method on three EEG datasets. Section 5 and Section 6 present and discuss the obtained results, respectively. Finally, we conclude this work with a summary of the advantages and limitations of the proposed approach in Section 7.

## 2. Shapley Value and Integrated Gradients

The SS method approximates the Shapley value, which was originally proposed to fairly assign gains to players in cooperative game theory [19]. The key idea of this approach is that a given player’s contribution to the result of a game is defined as the amount of gain reduction in his/her absence. The amount of gain reduction is averaged over all possible combinations of players and then normalized. For a machine-learning-based classification, the input features and classifier outputs of the models are respectively regarded as the players and the gain. This calculation has a computational complexity of O(n!) for *n* input features and retraining of the classifier for all possible combinations [20]. The SS method approximates the Shapley value using a Monte Carlo sampling method to avoid retraining and reduce computational costs [11,16]. Therefore, the reliability of the SS method depends on the number of samples available. The SS value is calculated for each data example, and the calculated values are averaged among the data to obtain the contributions of input features. However, this approximation requires many samplings, i.e., forward computations from input to output, and is repeated as many times as the number of data points.

The Shapley value is a unique method that satisfies the following four desirable properties for the reliable computation of contributions to a classification [11,13,15,16,21]:
ISymmetry (A): for any combinations of features *S*, if the change in the classifier output *f* is identical when a feature $x_{i}$ or $x_{j}$ is added, the features $x_{i}$ and $x_{j}$ contribute equally to the output: $f_{S\cup i}\left(x_{S\cup i}\right)=f_{S\cup j}\left(x_{S\cup j}\right)\to \phi_{i}\left(f,x\right)=\phi_{j}\left(f,x\right)$. Here, $\phi_{i}\left(f,x\right)$ indicates the amount of the contribution of the *i*-th feature in a classifier *f* and input features x for a data sample from a dataset X (x∈X). This property ensures that if two variables lead to identical outputs, the same contributions to the outputs are attributed to the variables.IIDummy: if a classifier does not depend on some variables, these variables always have zero contribution: $f_{S}\left(x_{S}\right)=f_{S\cup i}\to \phi_{i}\left(f,x\right)=0$. This property captures the desired insensitivity of the attributions.IIILinearity: let a classifier *f* be represented by the weighted linear sum of two subclassifiers $f_{1}$ and $f_{2}$; that is, $f=af_{1}+bf_{2}$ for scalars *a* and *b*. Then, the contribution of classifier ϕ(f) is also obtained by the weighted linear sum of the contribution of each subclassifier: $\phi \left(f\right)=a\phi \left(f_{1}\right)+b\phi \left(f_{2}\right)$. This property intuitively enables the method to fairly attribute the contributions into linearly combining submodules within the classifier.IVCompleteness: the sum of the contributions of all features equals the classifier’s output value: $\sum_{i=1}^{M}\phi_{i}\left(f,x\right)=f\left(x\right)$. Satisfying this property is desirable if the classifier’s output is continuous and used in a numeric sense.

The IG method is a type of path method [13,21] that integrates the gradients of the output with respect to the classifier’s input, usually before a softmax function in the output layer, along an arbitrary path from the baseline to the input data point. Given smooth path function $\gamma \left(\alpha \right)=\left[\gamma_{i}\left(\alpha \right),\ldots ,\gamma_{n}\left(\alpha \right)\right]$ for α∈[0,1] specifying a path from a baseline to the input data x and classifier *f*, the contribution of the *i*-th feature ${\mathrm{PathIG}}_{i}^{\gamma }\left(x\right)$ is given by
(1)$${\mathrm{PathIG}}_{i}^{\gamma }\left(x\right)=\int_{\alpha =0}^{1}\frac{\partial f\left(\gamma (\alpha )\right)}{\partial \gamma_{i}\left(\alpha \right)}\frac{\partial \gamma_{i}\left(\alpha \right)}{\partial \alpha }d\alpha .$$

The path method satisfies properties II–VI and the following additional property.

VImplementation invariance: if two classifiers f(x) and g(x) are functionally equivalent; that is, if they provide equal outputs for every input, the attribution of contributions to the outputs is also equal: f(x)=g(x)→ϕ(f,x)=ϕ(g,x).

If the path is a straight line:(2)$$\gamma \left(\alpha \right)=\tilde{x}+\alpha \left(x-\tilde{x}\right),$$
the method is referred to as IG. Here, $\tilde{x}$ is a baseline that is a non-feature input. The IG method further satisfies the symmetry (B) property defined below as well as the properties II–V mentioned above [13].

VISymmetry (B): assume that a classifier output does not always change even when exchanging features $x_{i}$ and $x_{j}$; that is, $f\left(x_{i},x_{j}\right)=f\left(x_{j},x_{i}\right)$. Then, the contributions of these features are identical: $\phi_{i}\left(f,x\right)=\phi_{j}\left(f,x\right)$.

The IG method is actually derived from properties II–VI. If the symmetry (B) property (VI) is satisfied, the symmetry (A) property (I) is also satisfied. Therefore, the IG method can be used to calculate the Shapley value. For the actual integral calculation, the integration interval is divided into many subintervals in which linear changes are assumed.

In most cases, a zero input is used as a baseline instead of a better input (see blue arrow in Figure 1A). However, a zero input is often inappropriate and results in the unreliable attribution of contributions to the results [17].

## 3. Compensation of Integrated Gradients

To compensate for the contributions to the classification from the user-defined zero (arbitrary) baseline, we calculate the difference between the contribution obtained using this baseline and a reliable contribution obtained using the SS method for a single datum example (orange dashed arrow in Figure 1A). Because it was proven that the Shapley value is a unique method satisfying the desirable properties [21], computing the SS is practically equivalent to computing IG with the true baseline that satisfies the desirable properties. The calculation algorithms differ, but the results should be equivalent. Note that the computational cost of this step is not very high, because the SS method is only applied to a single datum example. Negative operations when computing the difference imply the integral along the opposite arrows in Figure 1A. Therefore, this difference between the contributions obtained using the SS method and the contributions obtained using the IG method with the user-defined baseline corresponds to the integral from the appropriate baseline to the user-defined baseline through the given data point. Because the integral of the gradients along any path exhibits the same value, the difference finally corresponds to the integral of the gradients along an unknown path from the true baseline to the user-defined baseline. The integral value does not depend on the data as it is determined only by the two baselines. Therefore, the difference for one data example can be applied to the contributions to the classification obtained using the IG method with the baseline and the SS method for other data. The integral value is added to the contribution to the classification obtained using the user-defined baseline and into an arbitrary target data point to obtain the input and output gradients integrated along the orange path in Figure 1B.

The compensated IG is calculated as follows. First, a classifier is trained with a dataset X. Then, the contributions of the input features for a single datum example (x∈X) are obtained using the SS method. The contributions for the same data example x are obtained using the IG method with an arbitrary baseline $\tilde{x}$. The difference between these contributions for x is added to the contributions for the other data examples, which are obtained using the IG method with the baseline $\tilde{x}$. The compensated contributions are averaged among data examples.

The path method, including the compensated IG, satisfies properties II–V. We prove that the compensated IG method also satisfies the symmetry property (VI) if the classifier processes of the input features are independent and identical. Let an input time-series be *n*-dimensional: $x=[x_{1},\ldots ,x_{i},\ldots ,x_{n}]$; let the length of each time-series *z* be *L*: $x_{i}=\left[z_{i1},\ldots ,z_{ik},\ldots ,z_{iL}\right]$, and let a classifier f(x) be given by
(3)$$f\left(x\right)=\sum_{i}^{n}\sum_{j}^{m}W_{ij}g_{j}\left(x_{i}\right),$$
where $W_{ij}$ denotes a weight and $g_{j}\left(x_{i}\right)$ is a nonlinear function that converts time-series input $x_{i}$ into *m* outputs. Therefore, the input features are not combined when processing each $g_{j}$ (independent constraint), and each feature is processed by the same $g_{j}$ (identical constraint).

In an EEG signal classification using a CNN model in this study, $x_{i}$ indicates the time series measured by the *i*-th sensor or electrode, as shown in Figure 2. Assume that the contribution of $x_{i}$ to the CNN’s output is to be evaluated. For the independent constraint, $g_{j}\left(x_{i}\right)$ represents one-dimensional (temporal) convolutional layers or filters that do not spatially convolve the features. The processing $g_{j}\left(x_{i}\right)$ of each feature shares a common architecture with the same parameters (weights and biases); that is, for example, the process is realized by a shared-weight technique that is widely used in CNNs. $W_{ij}$ represents weights in the fully connected layer, as shown in Figure 2.

**Theorem** **1.**
*If a classifier is represented as Equation (Equation 3), all path methods always satisfy the symmetry (B) property (VI). Therefore, explanation methods satisfying Equation (Equation 3) and properties II–V always satisfy property (VI).*


**Proof.** The contribution of the *p*-th feature $x_{p}$ to the outputs is represented as $\phi_{p}\left(f,x\right)$. From Equation (Equation 3),
(4)$$\phi_{p}\left(f,x\right)=\phi \left(\sum_{i}^{n}\sum_{j}^{m}W_{ij}g_{j}\left(x_{i}\right),x\right).$$Using the linearity property (III) yields
(5)$$\phi_{p}\left(\sum_{i}^{n}\sum_{j}^{m}W_{ij}g_{j}\left(x_{i}\right),x\right)=\phi_{p}\left(\sum_{j}^{m}W_{pj}g_{j}\left(x_{p}\right),x\right).$$Using the implementation invariance property (V) and the symmetry assumption (let symmetric features be $x_{q}$) gives
(6)$$W_{pj}=W_{qj},$$
and
(7)$$g_{j}\left(x_{p}\right)=g_{j}\left(x_{q}\right).$$Therefore,
(8)$$\phi_{p}\left(\sum_{j}^{m}W_{pj}g_{j}\left(x_{p}\right),x\right)=\phi_{q}\left(\sum_{j}^{m}W_{qj}g_{j}\left(x_{q}\right),x\right).$$Then,
(9)$$\phi_{p}\left(f,x\right)=\phi_{q}\left(f,x\right).$$Consequently, all path methods satisfy symmetry property (VI) if the processes of input features are independent and identical in a given classifier. □

The contributions to the output of temporal CNNs satisfy all the properties because each filter convolves an input time-series feature; that is, they are represented as Equation (Equation 3). In contrast, the contributions to the output of spatial or spatiotemporal CNNs, which are commonly used in image recognition, do not satisfy the symmetry property (VI) because they violate the independent constraint and are not represented by Equation (Equation 3). Even if the explanation does not satisfy the symmetry property (VI) due to the proposed compensation, it might still be more reliable than that obtained from an inappropriately baselined IG method. As mentioned in the previous section, the path method, including the IG method, assuming an appropriate baseline, uniquely always satisfies the properties II–VI. Therefore, if the baseline is not appropriate, the method can violate the properties, and it has been empirically shown that the reliability of the resultant explanations is greatly impaired [17].

Satisfying these properties, II–VI is desirable in explaining EEG signal classification. The proposed method can attribute the same contributions to two electrodes that play the exact same role in the classification due to the symmetry property (VI). Owing to the dummy property (II), the method can ignore any insensitive electrodes. The linearity property (III) is desired to compute the contributions of linear combinations of CNN filters. The completeness property (IV) enables the method to calculate a large sum of the contributions if the output (i.e., classification confidence) is large.

## 4. Evaluation Methods

### 4.1. Datasets

We evaluated the reliability and computational costs of the proposed method on three publicly available EEG datasets, including the PhysioNet polysomnography (PSG) dataset [22], the UCI EEG dataset [23], and the CHB-MIT Scalp EEG dataset [24]. We fed the raw data into CNN models without any preprocessing.

The PhysioNet PSG dataset includes three channels: EEG channels from two electrodes and an electrooculography channel measured during sleep at a sampling frequency of 100 Hz. Data from 20 healthy people were collected for two days. A single epoch lasted 30 s, each of which was given a label describing sleep stages: non-rapid eye movement 1 (N1), N2, N3, rapid eye movement (R), or wakefulness (W). This dataset has been used as a benchmark to classify sleep states. Because only three sensors were involved, this dataset does not contain a great deal of spatial information.

In contrast, the UCI EEG dataset and CHB-MIT Scalp EEG dataset include data recorded from 64 and 22 electrodes, respectively, and therefore, may contain more spatial information than the PhysioNet PSG dataset. The UCI EEG dataset consists of visual event-related EEG signals for a single second; these were measured from 64 electrodes at a sampling frequency of 256 Hz. Three channels were excluded because they were not EEG signals; 122 were included, of which, 45 were healthy and 77 had alcoholism. The CHB-MIT Scalp EEG dataset includes 22-channel EEG signals measured from 22 patients with epilepsy. The sampling frequency was 256 Hz. The time and duration of epileptic seizures were annotated.

### 4.2. Implementation

For the PhysioNet PSG dataset, we trained six-layer temporal CNN models to classify the data into five sleep stages (N1, N2, N3, R, and W), which convolved input EEG signals separately from one another to acquire time domain features integrated into the fully connected output layer (see Figure 2). Moreover, 50, 100, 200, 400, and 200 temporally convolutional filters were used with time window sizes of 50, 50, 50, 50, and 1, respectively, from the input side. The temporal max-pooling sizes from the first to third layers were 5, 2, and 2, respectively, and temporal average pooling with a size of 150 was adopted in the fourth layer. The final layer was fully connected. Batch normalization was used in all layers except for the final layer. We used the cross-entropy loss function and trained the CNNs using the Adam optimizer, where the minibatch size was 50. The CNNs were trained using data for the first day, and then they were tested using data for the second day.

For both the UCI EEG and the CHB-MIT Scalp EEG datasets, we trained four-layer temporal CNNs and four-layer spatiotemporal CNNs to classify the data into two classes (alcoholism/control and seizure/non-seizure, respectively). Data from subjects whose last digit subject numbers were 0 or 1 were used to test the CNNs, and those of the other subjects were used to train the CNN models. In the temporal CNNs, the time window sizes were 20, 20, and 10, and the numbers of filters were 20, 40, and 10, respectively, from the input side. The first and second layers used max-pooling with a size of 4, and the third layer used average pooling with a size of the full length. The final layer was fully connected.

In the spatiotemporal CNNs, the first layer performed temporal convolution; the temporal window size was 30, the number of filters was 20, and the max-pooling size was 2 in the direction of the time axis. The second layer performed spatial convolution; the spatial window size was 64, the number of filters was 40, and the max-pooling size was 2. The third layer performed temporal convolution; the temporal window size was 30, and the number of filters was 40. Further, the final layer was fully connected. The other parameters were the same as those for the CNNs for the PhysioNet PSG dataset. The spatiotemporal CNNs were two-dimensional, which prevented the proposed method from satisfying the symmetry property (VI).

In each dataset, we randomly selected 200 data samples of each class. For each classifier, we computed the contributions of input features to the outputs using the proposed method, the IG method with zero-input baseline, and the SS method. We used 10 additional data examples with the proposed method to mitigate the sampling error in the SS method, although theoretically a single datum example should be enough for this purpose. Ideally, the contribution to the classification should be compared against true contributions; however, these were unknown. Therefore, we considered the contributions to the classification obtained using the complete SS method as the true contributions and measured the similarity among contributions in terms of Spearman’s correlation [25,26]. Large correlation coefficients (close to 1) indicate high similarity between the two compared contributions to the classification. Note that the comparison with the contributions obtained using the SS method reflects the sampling errors. Furthermore, we determined the contributions of EEG electrodes to the classification of alcoholism in the UCI EEG dataset to qualitatively compare the methods. The number of divisions of the integral interval in the IG methods was set to 200. The number of samplings in the SS method was set to 1000 per data example. We used a temporal model and averaged the contributions from the 200 data samples analyzed for this comparison.

## 5. Results

Table 1 lists Spearman’s correlation coefficients of all datasets and models. The coefficient values of the proposed compensated IG method (C-IG) were approximately equivalent to those of the SS method on the temporal CNNs. In contrast, the original IG method exhibited the lowest values, especially for the UCI EEG dataset. Therefore, the proposed method can suitably compensate for the unreliable contributions to the classification obtained using the IG method with the zero-point baseline.

Even when C-IG did not satisfy the symmetry property (VI) on the spatiotemporal CNNs, its coefficient values were larger than those of the original IG method with the zero-input baseline. However, the values obtained by C-IG were lower than those of the SS method in this case, indicating that the violation of the symmetry property (VI) undermined the reliability of the estimated contributions to the classification. Nonetheless, compensation improved the reliability of the original IG method.

We defined the computational complexity of the explanation methods in the UCI EEG dataset. The complexity of the original IG method was given by the number of divisions of integral intervals (200) × number of data examples (200) = 40,000. The proposed method exhibited additional complexity for preprocessing using SS, given by the number of samplings (1000) × number of electrodes (61) × number of data examples (10) = 610,000; therefore, its total computational complexity was 650,000. The complexity of the SS method was given by the number of samplings (1000) × number of electrodes (61) × number of data examples (200) = 12,200,000. Assuming equal computational costs for the approximated integrals (backward computation) and the samplings (forward computation), the computational complexity of the proposed method was less than that of the SS method by approximately 95%. When 1000 data examples were targeted, the complexity was 99% lower. Table 2 summarizes the computational complexity of the explanation methods considered.

Table 3 shows the classification accuracy. The performances of the spatiotemporal CNNs were better than those of the temporal CNNs. The constraints of the classifiers for more reliable explanations limited their learning performances.

Figure 3 shows the contributions of the EEG electrodes for alcoholism (A) and control (B) classification using the temporal CNNs in the UCI EEG dataset. They averaged over 100 randomly selected data. The contributions to the classification obtained using the proposed method (C-IG: left panels) were indistinguishable from the true contributions obtained using the SS method (middle panels), whereas those obtained using the original IG method (right panels) exhibited different distributions from those obtained using other methods. The correlations between the contributions obtained using the SS method and the proposed method are shown in the column of temporal CNNs (C-IG) in the row of UCI EEG in Table 1, indicating a high correlation of more than 0.99. In contrast, the contribution obtained using the original IG method (right panels) shows different distributions from those obtained using other methods.

A comparison of the contribution distributions in (A) and (B) reveals an inverse relationship in the proposed and SS methods; specifically, many areas showing positive contributions for the alcoholism class tended to show negative contributions to the healthy class, and vice versa. In contrast, the contribution distributions in (A) and (B) obtained using the original IG method were similar, indicating the risk of false explanation of the IG method with an inappropriate baseline and the importance of compensation.

Figure 4 shows the counterparts in the spatiotemporal CNNs. The contributions to the classification obtained using C-IG were slightly different from those obtained using the SS method. The correlations between the contributions obtained using the SS method and the C-IG method are shown in the column of spatiotemporal CNNs (C-IG) in the row of UCI EEG in Table 1, indicating approximately 0.7. These differences originated from the violation of the symmetry property (VI) in the spatiotemporal CNNs. Nonetheless, the compensation improved the explanation obtained by using the original IG method.

## 6. Discussion

In this work, we proposed a compensated IG method using one example of the SS value to improve the reliability of learning models in explaining classification outcomes. The proposed method satisfies the four desired properties, namely, the dummy, linearity, completeness, and implementation invariance properties, as well as the additional symmetry property under the classifier constraints. By using three kinds of EEG datasets, we demonstrated that the proposed method was able to compute more reliable contributions to the classification results than the original IG method (see Table 1) and with a much lower computational cost than the SS method. The contributions to the classification obtained using the proposed method were very similar to those obtained using the SS method, especially for temporal CNNs, and satisfy the constraints of the symmetry property.

However, the classifier constraints can reduce the classification accuracy (see Table 3). In contrast, the spatiotemporal CNNs showed higher classification accuracy but lower explanation reliability than the temporal CNNs. Indeed, spatial features, including brain functional connectivity, are important for EEG analyses [27,28], although the proposed method assumes that features are spatially independent. This constraint is a limitation of the method. Therefore, the classifier selection should depend on whether the explanation reliability or classification accuracy is emphasized. Even when the symmetry property is violated given the convolution among input features, we demonstrated that compensation effectively improves the reliability of the original IG method because the compensated IG can satisfy properties II–V.

Very little spatial information is included in the PhysioNet database. In this case, the proposed method was able to reliably compute contributions of data points over the time series, whereas classifiers were not affected by the spatial constraint. Recently, an easy-to-use EEG with a small number of channels was developed [29]. The proposed method can be applied to such EEG signals without spatial features, as shown in the results for the PhysioNet database.

The visualized contributions to alcoholism classification focused on the occipital visual area and frontal area (see Figure 3 and Figure 4). This was remarkably evident in the explanation using the proposed method and not observed in the explanation using the original IG method. The EEG signals in the UCI EEG dataset were measured in a visual-event-related design. Neuroscientific studies have found that the positive amplitudes to visual stimuli are significantly lower in people with alcoholism than in those without the condition [30,31]. In addition, a statistical analysis of the same dataset showed different patterns of EEG powers between participants with alcoholism and the control group in the occipital and frontal regions [32] and discussed a possible link to cortical atrophy [33]. Classifiers might use such differences to identify alcoholism, and the proposed method visualized the contributions of visual areas in the occipital and frontal lobes. However, additional analyses in the frequency domain and a coherence analysis between electrode signals are necessary to examine the plausibility of the visualizations in more details.

Our method evaluates the contributions of electrodes; that is, the spatial input space. However, for the classification of EEG signals, contributions in the frequency domain may be more interpretable than those in the spatiotemporal domain [4,7,18]. Many explanation methods, including the IG method, visualize the contributions in the input space because they were developed for image classification. Therefore, a more detailed explanation could be obtained by transforming the input contributions from the time domain to the frequency domain.

The visualization of the contributions of classifier variables (e.g., nodes) is important to understand the internal processes of the classifier, whereas we visualized the contributions of the outputs in this study. For example, the correlation between inputs and activities of a node [4] and information entropy of activities of a node [34] were used to clarify representation of each node in EEG signal classification. The proposed method can also be applied to the internal variables by considering the activities of a node as the outputs. This may offer more reliable explanations of the learning representation of each node than the existing methods. In future studies, we plan to apply the proposed method to multichannel time series data that differ from EEG data, such as the acceleration of human activities [35] and electrocardiography [36].

## 7. Conclusions

In this study, we proposed a method to compensate for the IG explanation method. The proposed approach is theoretically superior and has lower computational costs; it uses one SS value to solve the problem that the reliability of the explanation decreases when the baseline of the path integral is set inappropriately. The proposed method can compute highly reliable contributions to the classification regardless of the baseline if individual input features are non-linearly processed in an identical manner and the processed features are then linearly combined to create outputs that are often used for EEG signal classification. By using three different EEG datasets, we demonstrated that the proposed method was able to calculate contributions to the classifications that were as reliable as those obtained using the SS method. The computational complexity of the proposed method is almost the same as that of the conventional IG method and is much smaller than that of the SS method.

## Figures and Tables

**Figure 1 brainsci-12-00849-f001:**
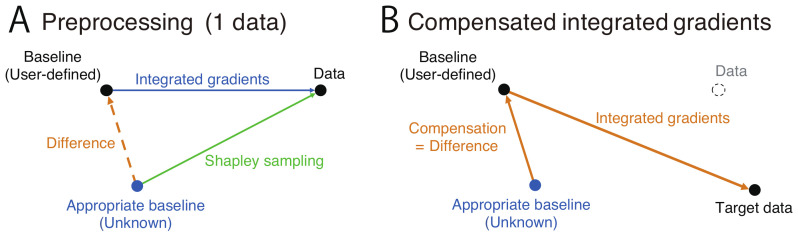
Diagram of the proposed method. (**A**) Preprocessing to compute the compensation amount (orange dashed arrow), given as the difference between the contribution to the classification obtained using the IG method with an arbitrary user-defined baseline (blue arrow) and the contribution to the classification obtained using the SS method that implicitly exhibits an appropriate baseline (green arrow). (**B**) Proposed IG method that compensates for the divergence from the appropriate baseline. This is equivalent to the path integral along the orange path.

**Figure 2 brainsci-12-00849-f002:**
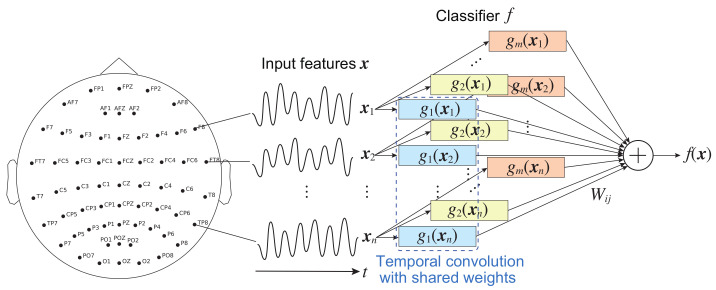
Model architecture for EEG signal classification using the CNN with constraints. Time-series signals measured by electrode $x_{i}$ for i=1,…,n are fed into temporal convolution layers $g_{j}\left(x_{i}\right)$ for j=1,…,m. The input features are not spatially convolved owing to the independent constraint. The processing $g_{j}\left(x_{i}\right)$ consists of the same parameterization owing to the identical constraint. If a classifier satisfies these constraints, the compensated IG can reliably compute the spatial contributions of $x_{i}$ to the outputs.

**Figure 3 brainsci-12-00849-f003:**
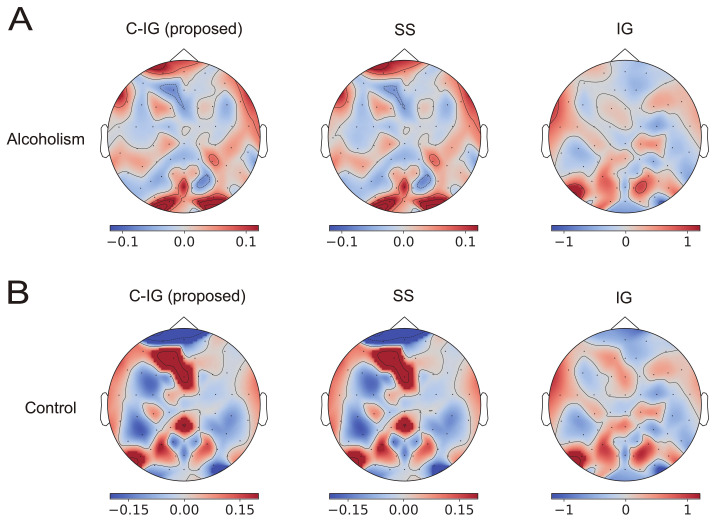
Contributions of EEG electrodes over the scalp to classify alcoholism (**A**) and control (**B**) using the *temporal* CNNs. The average contributions of the methods applied to 100 data examples from the UCI EEG dataset are shown. The red and blue areas represent positive and negative contributions to the classifications, respectively. The proposed method involves the compensated integrated gradients (left panels), the Shapley sampling shows the desired contributions as a reference (middle panels), and integrated gradients show the contributions obtained using the original integrated gradients method with the zero-input baseline (right panels).

**Figure 4 brainsci-12-00849-f004:**
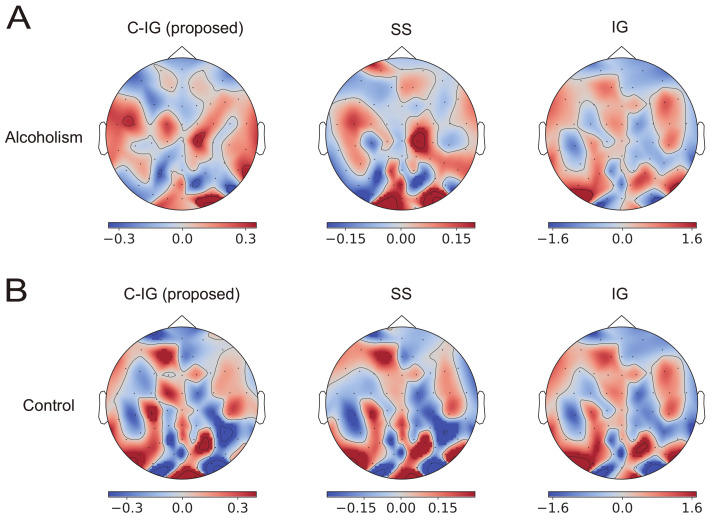
Contributions of EEG electrodes over the scalp to classify alcoholism (**A**) and control (**B**) using the *spatiotemporal* CNNs. The average contributions of the methods applied to 100 data examples from the UCI EEG dataset are shown. The red and blue areas represent positive and negative contributions to the classifications, respectively. The proposed compensated integrated gradients method is shown (left panels), while the Shapley sampling shows the desired contributions as a reference (middle panels), and integrated gradients show the contributions obtained using the original integrated gradients method with the zero-input baseline (right panels).

**Table 1 brainsci-12-00849-t001:** Spearman’s correlation coefficient compared to contributions obtained using the Shapley sampling. The temporal and spatiotemporal CNNs, respectively, correspond to one- and two-dimensional convolutional CNNs.

		Temporal CNNs			Spatiotemporal CNNs		
Dataset	Class	C-IG (Proposed)	SS	IG	C-IG	SS	IG
PhysioNet	N1	**0.983**	0.970	0.180			
	N2	**0.970**	0.953	0.655			
	N3	**0.988**	**0.988**	0.925			
	R	0.963	**0.970**	0.665			
	W	**0.987**	0.972	0.326			
CHB-MIT	Szr.	**0.996**	0.994	0.817	0.806	**0.983**	0.695
	No Szr.	**0.993**	0.990	0.293	0.917	**0.982**	−0.037
UCI EEG	Alc.	**0.994**	0.991	0.260	0.793	**0.989**	0.323
	Ctr.	**0.995**	0.992	0.258	0.739	**0.988**	0.331

C-IG, compensated integrated gradients (proposed method); SS, Shapley sampling; IG, integrated gradients with
the zero-input baseline; N1–3, non-rapid eye movement 1–3; R, rapid eye movement;W, wakefulness; Szr., seizure;
No Szr., non-seizure; Alc., alcoholism; Ctr., control. The numbers in bold indicate the best performance of the
C-IG, SS and IG methods.

**Table 2 brainsci-12-00849-t002:** Computational complexity of explanation methods in the UCI EEG dataset.

Method	Computational Complexity
IG	40,000
SS	12,200,000
C-IG	650,000

IG, integrated gradients with the zero-input baseline; SS, Shapley sampling; C-IG, compensated integrated
gradients (proposed method).

**Table 3 brainsci-12-00849-t003:** Classification accuracy of models for three datasets.

Dataset	Model	Accuracy
PhysioNet	Temporal CNNs	81.6%
CHB-MIT	Temporal CNNs	83.4%
	Spatiotemporal CNNs	88.7%
UCI EEG	Temporal CNNs	69.5%
	Spatiotemporal CNNs	77.1%

## Data Availability

The PSG dataset (version 1) is available at https://www.physionet.org/physiobank/database/sleep-edfx/ accessed on 24 June 2022 in PhysioNet. The CHB-MIT dataset is available at https://www.physionet.org/content/chbmit/1.0.0/ accessed on 24 June 2022 in PhysioNet. The UCI EEG dataset is available at https://archive.ics.uci.edu/ml/datasets/eeg+database accessed on 24 June 2022 in the UCI Machine Learning Repository.

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
