# Peer review of "Compensated Integrated Gradients for Reliable Explanation of Electroencephalogram Signal Classification"

_brainsci, 2022, doi:10.3390/brainsci12070849_

Round 1

Reviewer 1 Report

The Authors propose a method to compensate for the baseline selection of the IG method. The proposed method is well described, demonstrated mathematically, and evaluated with three publicly available EEG databases. Results demonstrate superior performance compared to the  IG method at a lower computational cost than the SS method. The paper is easy to follow, the literature review seems thorough, and the results are sound. Here are a few comments for the Authors consideration. 

  1. The method assumes spatially independent features. However, there are several features, mostly related to brain functional connectivity, that are equally important for analyzing the EEG data. I suggest the Authors discuss this point or list it as one of the potential limitations. 
  2. One of the datasets considered for the evaluation of the proposed method does not provided proper representation of EEG data (PhysioNet) as it involves only two EEG electrodes and lacks any spatial information about brain activities. 
  3. In the results section, the Authors state that the contributions to the classification obtained using the proposed C-IG method were indistinguishable from the true contributions obtained using the SS method. By looking at Figures 3 and 4, there are clear differences (left and middle panels), for example in the midfrontal area. It is not clear what indistinguishable mean here, and how it was measured. 
  4. It would also be useful to cross-validate the results with neuroscience findings, are these results in line with previous neuroscience research about the specific classification, i.e. in relation to alcoholism? 

I found a typo in line 7: can computes -> can compute. 

Author Response

The Authors propose a method to compensate for the baseline selection of the IG method. The proposed method is well described, demonstrated mathematically, and evaluated with three publicly available EEG databases. Results demonstrate superior performance compared to the IG method at a lower computational cost than the SS method. The paper is easy to follow, the literature review seems thorough, and the results are sound. Here are a few comments for the Authors consideration.

  1. The method assumes spatially independent features. However, there are several features, mostly related to brain functional connectivity, that are equally important for analyzing the EEG data. I suggest the Authors discuss this point or list it as one of the potential limitations.

[Authors’ response] Thank you for your advice. Indeed, spatial features including brain functional connectivity are important for EEG analyses [1, 2] although the proposed method assumes that features are spatially independent. This constraint is a limitation of the method.

We added this description to section 6 (Lines 336-339)

  1. One of the datasets considered for the evaluation of the proposed method does not provide proper representation of EEG data (PhysioNet) as it involves only two EEG electrodes and lacks any spatial information about brain activities.

[Authors’ response] That’s right. Very little spatial information is included in the PhysioNet Database. In this case, the proposed method was able to reliably compute contributions of data points over time series, whereas classifiers were not affected by the spatial constraint. Recently, an easy-to-use EEG with a small number of channels has been developed [3]. The proposed method can be applied to such EEG signals without spatial features, as shown in the results for the PhysioNet database.

We added this description to section 6 (Lines 344-349)

  1. In the results section, the Authors state that the contributions to the classification obtained using the proposed C-IG method were indistinguishable from the true contributions obtained using the SS method. By looking at Figures 3 and 4, there are clear differences (left and middle panels), for example in the midfrontal area. It is not clear what indistinguishable mean here, and how it was measured.

[Authors’ response] Table 1 shows the correlation coefficients between the contributions obtained using the proposed C-IG method and true contributions obtained using the SS method. The values corresponding to Fig. 3 are shown in the column of temporal CNNs (C-IG) in the row of UCI EEG, indicating a high correlation of more than 0.99. The values corresponding to Fig. 4 are shown in the column of spatiotemporal CNNs in the row of UCI EEG, indicating approximately 0.7. These figures reflect the correlations, and in Fig. 3, the results for C-IG and SS are almost identical, while in Fig. 4, the results are slightly different.

We added the explanations to section 5 (Lines 302-304 and 316-318)

  1. It would also be useful to cross-validate the results with neuroscience findings, are these results in line with previous neuroscience research about the specific classification, i.e. in relation to alcoholism?

[Authors’ response] The EEG signals in the UCI EEG dataset were measured in a visual-event-related design. Neuroscientific studies have found that the positive amplitudes to visual stimuli are significantly lower in people with alcoholism than in those without the condition [4, 5]. In addition, a statistical analysis of the same dataset showed different patterns of EEG powers between participants with alcoholism and the control group in the occipital and frontal regions [6]. Classifiers might use such differences to identify alcoholism, and the proposed method visualized the contribution of visual areas in the occipital and frontal lobes.

We added this discussion to section 6 (Lines 353-361)

I found a typo in line 7: can computes -> can compute.

[Authors’ response] Thank you pointing it out. We have done the English proofreading again.

[1] Sakkalis, “Review of advanced techniques for the estimation of brain connectivity measured with EEG/MEG,” Computers in Biology and Medicine, Vol. 41, No. 12, pp. 1110-1117, 2011.

[2] Cao et al., “Brain functional and effective connectivity based on electroencephalography recordings: A review,” Human Brain Mapping, Vol. 43, No. 2, pp. 860-879, 2022.

[3] Yoshimoto et al., “Wireless EEG patch sensor on forehead using on-demand stretchable electrode sheet and electrode-tissue impedance scanner,” in Proceedings of the 38th Annual International Conference of the IEEE Engineering in Medicine and Biology Society, pp. 6287-6289, 2016.

[4] Porjesz and Begleiter, “Effects of alcohol on electrophysiological activity of the brain,” In: Begleiter H, Kissin B, editors. Alcohol and Alcoholism, The Pharmacology of Alcohol and Alcohol Dependence. New York: Oxford University Press, pp. 207–247, 1996.

[5] Porjesz and Begleiter, “Genetic basis of the event-related potentials and their relationship to alcoholism and alcohol use,” Journal of Clinical Neurophysiology, Vol. 15, No. 1, pp. 44–57, 1998.

[6] Tcheslavski and Gonen, “Alcoholism-related alternations in spectrum, coherence, and phase synchrony of topical electroencephalogram,” Computers in Biology and Medicine, Vol. 42, pp. 394–401, 2012.

Reviewer 2 Report

The paper presents an interesting idea, which has also been supported with experiments. However, some concerns need to be addressed:

1. Grammatical and spelling errors to be corrected.

2. Justify the statement: "This difference between the contributions... ...corresponds to the integral of the gradients along an unknown path." (Lines 141-143).

3.The paragraph in Lines 175-185 lacks clarity of presentation, and should be rewritten. What do the authors aim to convey?

4. The section 5, "Evaluation Methods" gives a description of data as well as the models, but in an interspersed manner which is difficult to comprehend. There should be two subsections - one  for Dataset description, and the other for Implementation details.

5. In the Results section, Computational complexity has been defined and computed, however, it would be better to highlight it as a Table to make it more visible. This is all the more important since authors claim it to be an important contribution.

Author Response

The paper presents an interesting idea, which has also been supported with experiments. However, some concerns need to be addressed:

  1. Grammatical and spelling errors to be corrected.

[Authors’ response] Thank you pointing it out. We did the English proofreading.

  1. Justify the statement: "This difference between the contributions... ...corresponds to the integral of the gradients along an unknown path." (Lines 141-143).

[Authors’ response] Thank you for advice.

Negative operations when computing the difference imply the integral along the opposite arrows in Fig. 1. Therefore, this difference between the contributions obtained using SS method and the contributions obtained using the IG method with the user-defined baseline corresponds to the integral from the appropriate baseline to user-defined baseline through the given data point. Because the integral of the gradients along any path exhibits the same value, the difference finally corresponds to the integral of the gradients along an unknown path from the true baseline to the user-defined baseline.

We added the explanation to section 3 (Lines 140-147)

  1. The paragraph in Lines 175-185 lacks clarity of presentation, and should be rewritten. What do the authors aim to convey?

[Authors’ response] Thank you for your advice. We rewrote the text as follows (Lines 178-189).

The contributions to the output of temporal CNNs satisfy all the properties because each filter convolves an input time-series feature, that is, they are represented as Equation (3). In contrast, the contributions to the output of spatial or spatiotemporal CNNs, which are commonly used in image recognition, do not satisfy the symmetry property (VI) because they violate the independent constraint and are not represented by Equation (3). Even if the explanation does not satisfy the symmetry property (VI) due to the proposed compensation, it might still be more reliable than the inappropriately baselined IG method. As mentioned in the previous section, the path method including the IG method assuming an appropriate baseline uniquely always satisfies the properties II-VI. Therefore, if the baseline is not appropriate, the method can violate the properties, and it has been empirically shown that the reliability of the explanations is greatly impaired.

  1. The section 5, "Evaluation Methods" gives a description of data as well as the models, but in an interspersed manner which is difficult to comprehend. There should be two subsections - one for Dataset description, and the other for Implementation details.

[Authors’ response] Thank you for your suggestion. We changed section 4 to include two sections, one describing the dataset and the other describing how to implement the CNNs.

  1. In the Results section, Computational complexity has been defined and computed, however, it would be better to highlight it as a Table to make it more visible. This is all the more important since authors claim it to be an important contribution.

[Authors’ response] Thank you for your advice. We added a new table showing the computational complexity to section 5.